# The Contribution of Typological Studies to the Integrated Rehabilitation of Traditional Buildings: Heritage Enhancement of Urban Centres in Inner Alentejo, Portugal

Ana C. Rosado [1,2,*] and Miguel Reimão Costa [2,3]

¹ Dinâmia-CET, ISCTE, 1649-026 Lisbon, Portugal
² CEAACP, Campo Arqueológico de Mértola, 7750-353 Mértola, Portugal; mrcosta@ualg.pt
³ CEAACP, Universidade do Algarve, 8005-139 Faro, Portugal
* Correspondence: ana.rosado@iscte-iul.pt

**Abstract:** The urban centres of inland Alentejo (southern Portugal) have long faced a depopulation crisis which, besides undermining the cohesion of the communities, compromises the conservation of the architectural heritage. The tendency to apply the discourses on tourism and population pressures from coastal cities to these inland territories can be detrimental to their analysis given the sheer difference in demographic dynamics. Transformations in traditional architecture—a key facet for defining these historic urban landscapes—require analysing within this social context. The imperative need to rehabilitate traditional buildings, endowing them with the living conditions communities today require, must be guided by morphological analysis, knowledge of housing history, typologies, and traditional construction techniques. This rehabilitation concept integrates into the transformation processes that traditional architecture has been undergoing for centuries, constituting adaptable and flexible structures across their organisational variants, which should be studied through a prospective approach. The article characterises the transformation of urban domestic architecture in the region, from the early modern period to the present. The results of various research projects are summarised, gathering over 500 cases. The conclusion argues that the historical process itself results in a set of themes, tools, and opportunities for these buildings' adaptation to current needs.

**Keywords:** inland cities; demographic decline; urban decay; rehabilitation; typological study; adaptation; historical processes; heritage values

## 1. Introduction

Urban centres across the Alentejo have long been experiencing significant changes which, in most cases, were not accompanied by any registering of the typo-morphological process and constructive characteristics of the pre-existing architecture or the transformations taking place over time. These changes have emerged against a backdrop of regional depopulation in keeping with the reality prevailing across all inland Portugal. In urban contexts, depopulation reflects in the separation of historic areas (often contained within intramural spaces) from urban areas of (more or less) recent expansion, favoured by the resident population. The population drain of historic centres represents a by-product of both the advancing demographic abandonment and the frequent rejection of traditional housing typologies. Traditional housing is often perceived as failing in terms of the living conditions provided, which differ from those associated with current living standards. The abandonment of historic centres and the rejection of the traditional house thus pose a threat to the integrated conservation of the historic built ensembles. Therefore, any policy seeking to halt or reverse the process of demographic shrinkage must place conservation at the centre of the development model, considering its importance as a resource holding the potential to bring about the necessary reversal in the negative overlook prevailing in local communities. Identifying and sharing exemplary architectural rehabilitation projects might

make a fundamental contribution to this context. Likewise, it is important to record the fundamental characteristics of the traditional architecture of these nuclei and understand their diversity in the historical context in order to then recognise how certain of these characteristics—both spatial and constructive—are conducive to their own transformation and rehabilitation.

The adaptation of housing to new standards of comfort, needed to fix inhabitants and even newcomers, benefits from following the historic process of transformation as their flexibility allows for continuity, as will be presented in part 3. In this context, the presence of sustainable tourism—especially cultural tourism –remains a positive factor [1] impacting the demographic and heritage conservation of medium and small inland towns and cities. As defined by the United Nations World Tourism Organization, sustainable tourism accounts for current and future economic, social, and environmental impacts [2] while cultural tourism is an activity of learning, discovering, experiencing, and consuming the tangible and intangible cultural attractions/products in a tourism destination [3]. These concepts relate to local cultures' lifestyles, value systems, beliefs, and traditions, and engage in social and economic dynamics. However, the tendency to apply the discourses on tourism, gentrification and population pressures appropriate to large, coastal cities to these inland territories emerges as detrimental both to their specific analysis and their development [4]. The currently ongoing abandoning of urban centres only reiterates the importance of proximity policies, the participation of various actors seeking to integrate research and knowledge about the built environment, public housing policies and as well as private sector initiatives [5].

## 2. Materials and Methods

This article arises out of two research projects carried out in the inland Alentejo region between 2014 and 2022. The first project, "Traditional Architecture in Mértola's Old Town and its Territory: Built Heritage and Cultural Tourism", was financed by InAlentejo—CCDRAlentejo (Portugal) and focused on the town of Mértola, southern Alentejo, between 2014 and 2016. The second and subsequent project "Traditional Urban Housing in the Alentejo Region" was financed by the Portuguese Fundação para a Ciência e Tecnologia (FCT) and replicated the initial project's methodology over the larger scope of cities in the inner Alentejo: Estremoz, Borba, Moura and Serpa, during the years of 2017 and 2022 (Figure 1).

Both studies spanned the traditional housing types through a combined methodology of *in loco* architectural and photographic surveys of houses with archive information and inhabitant testimonies. A total of 77 houses were surveyed in Mértola before another 507 cases were gathered from the other four Alentejan cities—Estremoz, Borba, Moura and Serpa—of which 313 were derived from direct architectonic survey and 194 from the private construction registers maintained by municipalities—'processos de obras'. These are the building permit records that include a description of the works, the property's location, a set of technical drawings detailing the existing situation and a second set with the proposed alterations. This information allows for a complete understanding of the constructive characteristics of buildings alongside recent alterations and, therefore, to establish their equivalence with the surveyed buildings.

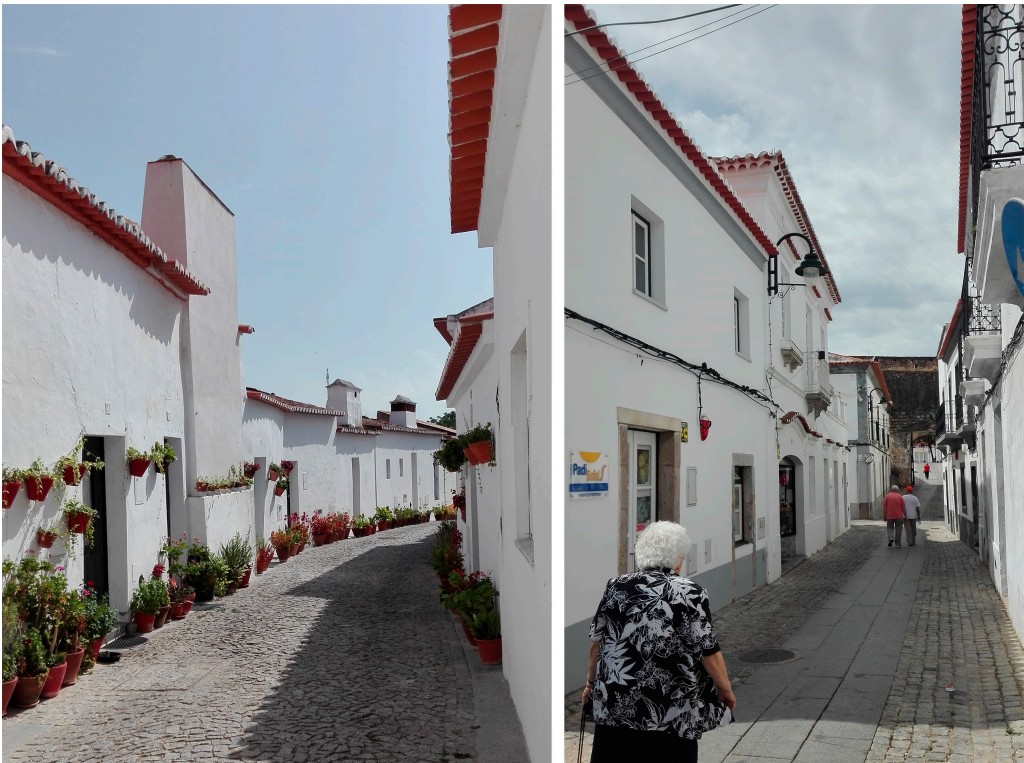

**Figure 1.** Streets in Moura and Serpa in 2018. An overlook of the studied cities' traditional urban architecture. Source: authors, 2023.

Additionally, the projects also gathered historical information on housing characteristics from the property record archives of the "Santa Casa da Misericórdia" religious organization in Serpa for 1673 (Tombo 1—SCMS, 1673) [6], the municipal estates of Estremoz, for 1674 (Tombo dos Foros 1674) [7] and 1746–1761 (Livro dos Tombos 1746–1761) [8], property acquisitions in Borba between 1597 and 1883 (Livros de Notas—CNB 1597–1883) [9] and the tax records of Mértola for 1765 (Livro da Décima—AMM 1765) [10]. This historical information enabled us to consistently portray the evolution in Alentejo's traditional housing models since the late 1600s. Moreover, inhabitant interviews provided information regarding alterations ongoing during the second half of the 20th century, including changes in the usages and functions of different parts of the houses and the nomenclature used and/or disappeared.

The geographic distribution of the case study cities encompasses the full extent of Alentejo's latitude while sharing the same territorial status as frontier cities, anchored around the 7°32′ O longitude (Figure 2). This distribution ensures the representation of almost all the inland Alentejo region through a diversified sample of case studies. All these settlements align with the central place definition proposed by Gaspar [11] as places holding central territorial functions, providing their hinterlands with goods and services. Proximity to the Spanish border drove a similar urban evolution based on the historical need to fortify both these towns and the territory that long persisted under an almost permanent menace of conflict. They correspondingly display similar urban features, such as the hilltop castles or citadels and successive lines of fortifications with their urban expansion strategies also reflecting clear affinities.

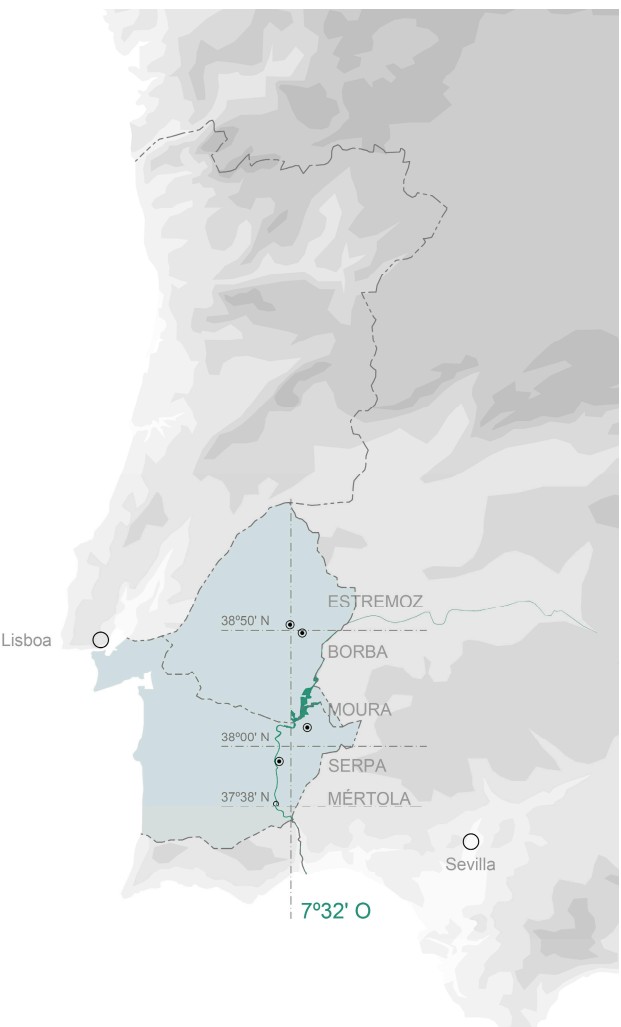

**Figure 2.** Case study locations inside the Alentejo region of Portugal. Source: authors, 2023.

## 3. Results

*Housing Architecture in Urban Alentejo: A Synthesis of Cases from Estremoz, Borba, Moura, Serpa and Mértola*

Most towns in the Alentejo interior share characteristics in terms of urban design, layout and landscape organisation that closely interlink with their frontier status. The historic centres commonly occupy high, strategic locations, confined within walls—perimeters almost always of medieval origin and often modernised or enlarged in the 17th century— many of which maintain the gates in the wall as the only access route [12]. As a rule, the urban fabric follows the guidelines of Portuguese medieval planning—particularly in the founding nuclei, but also in the transformation of pre-existing nuclei—with linear axes delimiting regular plots that are deeper than they are wider [13]. There is also widespread urban expansion into the surrounding lower areas, where another urban design model developed free from physical constraints. In general, there are no expressive processes of discrimination within these urban complexes, with a mixture of administrative, religious, and residential buildings; and within these, common and erudite houses appear side by side. Throughout history, domestic architecture has combined a series of distinct models associated with different typologies, scales and construction processes and a dimension of constant adaptability to changing circumstances.

The most common housing type in Inner Alentejo urban centers is in the adjoined rooms style, described as such in contrast with the medieval-Islamic house with its inner courtyard with surrounding rooms. The origin of this housing type derives from dividing

the two similarly sized squares that define the minimum dwelling unit and the basic solution for traditional houses throughout the region [14]. This also represents the basis from which more complex types arise in keeping with the dynamics of expansion, both in area and height and in the agglutination of neighbouring houses. While the basic two-room solution is associated with the medieval deep-and-narrow plot, its persistence over time reaches back beyond the regular medieval allotment. The two compartments encompass every basic housing need: one room for the family to gather, next to the street door, and a back room for sleeping and the storage of goods. The front room contained the activities related to preparing and cooking food and the main street door was frequently the only opening for lighting and ventilating the room.

The first transformation in this model involves expanding its depth through constructing a new cell (room) over a backyard that generally maintains the same constructive structure of load-bearing walls of stone masonry or rammed earth, approximately 60 cm thick, traditionally plastered and painted with lime. Even when the expansion is achieved through raising the height, with the addition of upper-level compartments, there are no significant constructive changes. These processes of house growth, both in-depth and height, are closely interlinked with the adaptation of the house to topographical conditions. Thus, it is not uncommon to encounter half-storey systems and houses structured in staggered levels, as seen in Estremoz (Figure 3) or Mértola (Figure 4), for example.

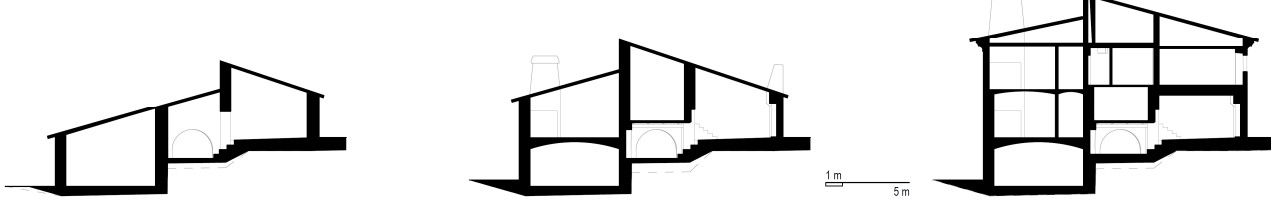

**Figure 3.** Section showing the evolution of the house on Rua Magalhães Lima no. 77–79, Estremoz. Source: Ana Costa Rosado, 2022.

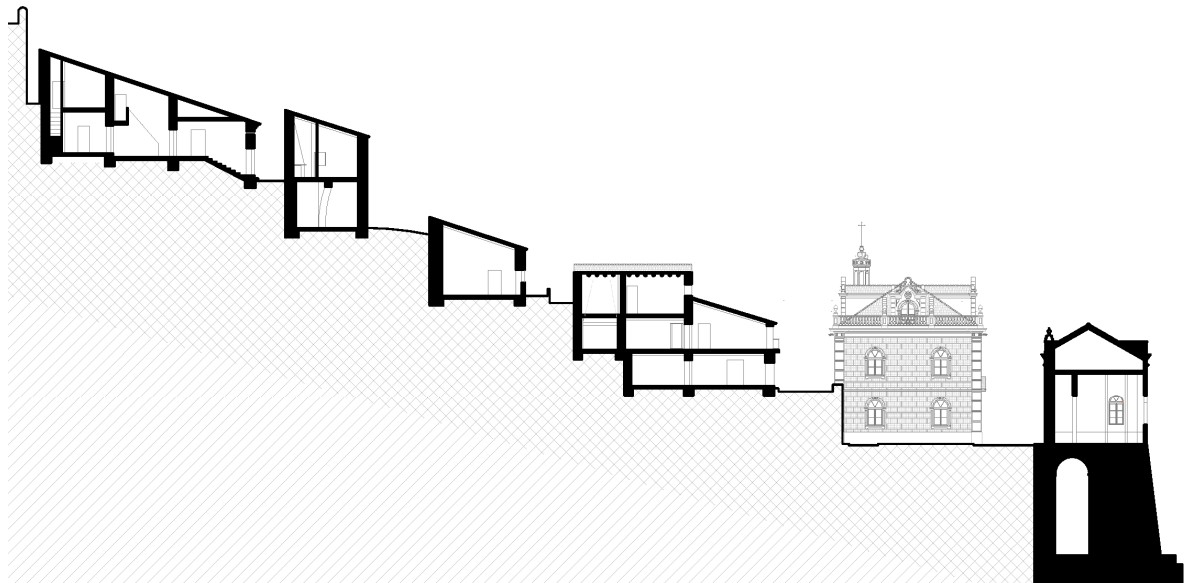

**Figure 4.** Section of Mértola, showing Praça Luís de Camões. The steep terrain leads to housing access on different floors. Source: authors, 2015.

In houses with upper floors, the spatial organisation continues to depend on the prevalence of solid load-bearing walls, with the flooring and roof timbers made up of purlins and rafters, and usually with only one flight of staircases, preferably leaning against

one of the side gables and not necessarily aligned with the main entrance. However, at a constructive level, this generic description spans significant diatopic diversity not only in relation to the solid walls (either stone masonry or rammed earth walls depending on the region) but also the floors themselves, with tiles [15] over beams very common in northern areas (as in Borba) and wooden flooring very prevalent in southern areas (as in Mértola).

The consolidation of these housing expansion and transformation processes resulted in more complex solutions, with both single-family and multi-family dwellings. The latter often combine several plots of land, and, in functional terms, progressively separate the commercial, storage and production spaces on the ground floor—shops, warehouses, cellars, etc.—from the family living spaces on the upper floors. The differentiation of dwellings by social class does not necessarily reflect in different construction systems but rather more evidently in larger scale buildings (due to the agglutination of more plots) and stylistic refinements to the first floor and façade (through installing balcony windows, pilasters, cornices and cymatiaor stone mouldings decorated to the tastes of each period) [16]. In any case, it is especially the multi-family typology that displays an important constructive transformation: recourse to solid brick for the construction of structural features—arches, vaults, and timbrel vaults—incorporated into domestic architecture through different solutions. The inclusion of arches in the solid ground floor walls, to obtain a larger surface area for commercial spaces, is of transversal importance throughout the region (Figure 5F,H). The use of vaults becomes frequent in Central Alentejo (for example in Estremoz and Borba- see [17]) and is almost generalised on the left bank of the Guadiana (Moura and Serpa). In residential areas, timbrel vaults (structures in which bricks are laid with their wider side facing the vault's surface) were more common as both a lighter and more economical solution.

The progressive opening of ground floor layouts for commerce and related activities brought about another important functional change: the kitchen tended to relocate to the upper floor and/or the back of the house. In smaller high-rise houses, when the kitchen is installed on the first floor, above the front room, the chimney body appears on the façade at the level of the upper floor (an architectural feature common in the latitude of Estremoz and Borba). In other cases, the purpose of opening up larger openings for lighting and ventilation resulted in the chimney body being removed from the façade and the kitchen being located at the back, with or without a connection to a backyard (Figure 5D,E). Similarly, in the erudite housing types, the positioning of the kitchen at the back also responded to the objectives of linking the main façade to the social spaces par excellence and ensuring the regular composition of the balcony openings.

Beyond the adjoining room housing type, in which all the compartments are delimited by load-bearing masonry or rammed earth walls, Inner Alentejo also features other housing solutions. A different model arises from progressive liberation from "heavy" construction systems combined with growing compartment specialisation and a greater preponderance of "light" construction systems. In these cases, the load-bearing structures are restricted to the exterior, roof supporting walls, and light partition walls (adobe, tiles, solid bricks, etc.) serving to organise the living spaces (Figure 5G,I). The importance of this model is clear: in the transformation of pre-existing buildings (especially when introducing a corridor and dividing some of the old compartments—Figure 5J); in the emergence of transitional models (Figure 5K); or, above all, in the construction of new homes on different scales (whether demolishing pre-existing buildings or occupying plots and parcels in areas undergoing urban expansion—Figure 5G). This change is accompanied by the growing importance of decorative expressions, both externally and internally, that acquires a diatopic dimension, as demonstrated, for example, by walls with paintings and carvings and the vaulted plasterwork decorated ceilings.

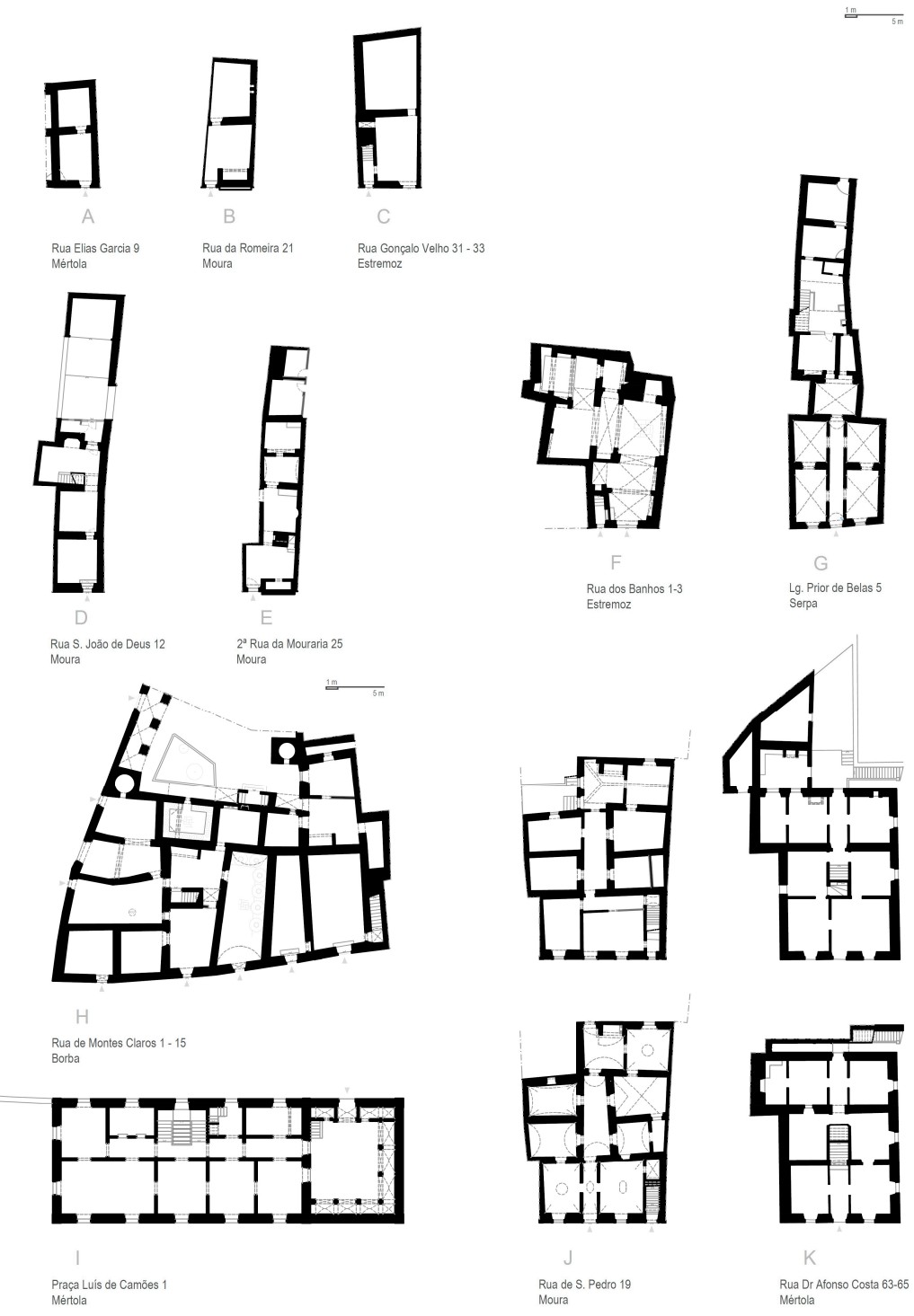

**Figure 5.** (**A**–**C**)—Houses with two rooms (**D**,**E**)—Two houses in Moura, of the same type, with differences in kitchen location—on the back, next to a yard and in the front room with a large chimney. (**F**)—Winery in Estremoz. (**G**)—House from the 19th–20th-century transition in an area of urban expansion in Serpa. (**H**)—House with independent units and winery in Borba. (**I**)—House from the 19th–20th-century transition following the demolition of pre-existing buildings in Mértola. (**J**,**K**)—Houses with a combination of load-bearing and partition walls around central corridors in Moura and Mértola. The former results from the transformation of an existing building while the latter was built from the ground up. Source: authors, 2023.

The gradual transformation of construction processes and the advent of industrially produced materials, especially from the middle of the 20th century onwards, saw the introduction of reinforced concrete to traditional buildings, especially in replacing the structural supports for floors and roofs. Pre-existing timbers, which are light and vulnerable to pathologies, have often been replaced by heavier concrete beams or slabs, often without this increase in load being taken into account in terms of the building's structural integrity. Nevertheless, the contemporary transformation of old buildings in the urban centres studied occurs within a tradition of change that cuts across all the housing models presented, which also sometimes includes altering the building structure. The dynamics of agglutinating contiguous plots or even incorporating neighbouring compartments are frequent procedures aimed at increasing the size of dwellings. Likewise, these interventions are often reversed by parcelling out existing buildings, thus closing doors and arches, both on the ground and upper floors. The versatility of this system (in which the primacy of the cellular composition remains despite everything) endows flexibility on the built fabric enabling constant adaptation in response to changes in society and the family structures themselves [18].

## 4. Discussion and Conclusions

### 4.1. Themes of Transformation from a Rehabilitation Perspective

The study of common housing in Portugal has largely focused on the rural landscape, almost always favouring the identification of the various regional types over the historical process or any diachronic reading. In general, urban centres lack an analytical document capable of framing an intervention in a specific building and evaluating its importance in the context of the history of domestic architecture. This circumstance contributes to the frequent option of complete or almost complete demolition of buildings whose state of conservation does not otherwise require this solution with such negative impacts on both heritage and the environment [19]. Nevertheless, the demolition of residential buildings (sometimes combined with alterations to allotment and property delimitations) to integrate another model of housing organisation was also very relevant in other periods of history. In addition to the best known and most expressive example of the transition from the courtyard house of the Islamic medieval period to the adjoining rooms house of the Christian medieval period, we may also include the consolidation of the bourgeois house model from the second half of the 19th century onwards which, in many cases, involved the demolition of several pre-existing buildings in the oldest centres, as happened in Mértola [20] (Figure 6).

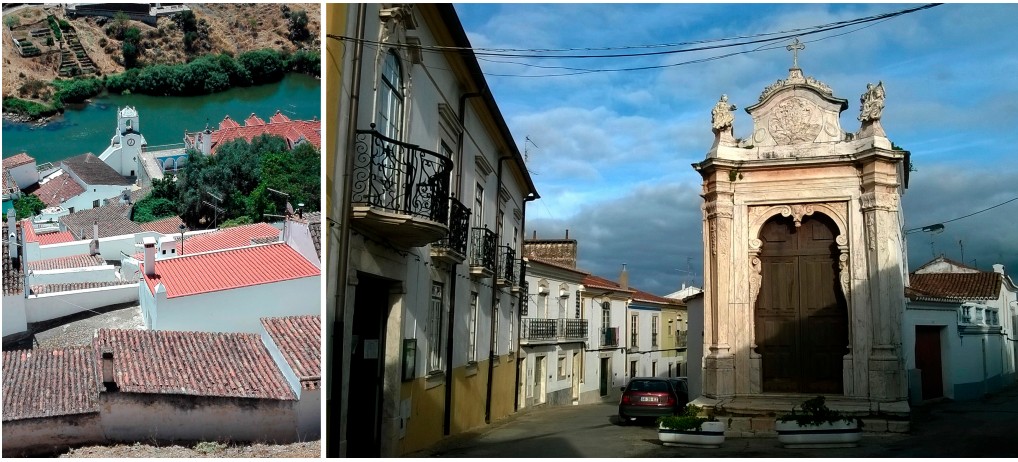

**Figure 6.** Mértola in 2020 and Borba in 2016. An overlook on studied cities' urban morphology. Source: authors, 2023.

As set out above, the study of traditional architecture in urban spaces reveals another pattern of intervention based on the conservation and successive transformation of

pre-existing buildings, seeking to respond to changing social, economic, and historical circumstances. This pattern incorporates a set of intervention practices that might be applied in contemporary rehabilitation projects conveying, in a certain sense, the most essential and timeless dimension of architecture, transversal to the most diverse typologies, chronologies and geographies. In its most elementary expression, this process might underpin the reorganisation of the articulation, linkage, and functionality of the various spaces in the house by closing and reopening old, boarded-up doors. As seen, this constitutes a recurring solution sometimes associated with alterations to the building structure, whether by combining adjoining plots or conversely by parcelling out larger dwellings as was recently studied in a building complex in Borba (Figure 5H).

The construction or removal of light walls reflects another recurring intervention in the history of domestic architecture that often translates into an expressive change in the way of living without compromising the building's structural concept. In this case, the aim involves altering the scale of the rooms in pre-existing buildings in order to achieve greater specialisation and privacy. This was the most common strategy employed from the mid-19th century onwards, with the compartmentalisation or introduction of the corridor into the larger dwelling spaces (installing partitions and other dividing walls—Figure 5J,K) in a set of interventions that may either be adopted or reversed by rehabilitation projects.

One of the most characteristic transformations of housing architecture in an urban context is the growth in height which, in traditional architecture, resulted in the frequent raising of the geometry of load-bearing walls. Similarly, any increase in the surface area and volume of buildings located in urban centres only seems justified today in cases of overly small dwellings. This solution, which in any case should always be included in detailed planning instruments, involves the introduction of new materials and construction processes that display compatibility with pre-existing structures, which has also happened at other times in history.

It is within this framework that scientific research takes on the greatest importance, focusing first and foremost on traditional building materials and techniques (and how their relevance can or cannot be converted to current production and market conditions), but also on newly compatible materials and techniques (trying to avoid the drawbacks of the "trial and error" evolution characteristic of vernacular culture). The complexity of some traditional structures means that technical training and the transmission of knowledge and skills from traditional techniques are fundamental to rehabilitation processes. Thus, surveying and cataloguing the construction (masonry, carpentry, etc.) know-how with the help of the professionals themselves should be carried out either in a "study and survey approach" or in an "applied approach" [21]. Likewise, the study and reconstruction of the different types of contemporary architecture and their transformation over time is also of great importance: not only with a view to their valuation and valorisation by the communities (essential for their conservation) [22], but also to broaden the range of solutions for rehabilitation which, in essence, falls within the tradition of successive adaptations of old buildings to new ways of living.

### 4.2. Rehabilitation in the Context of Inland Demographic and Tourist Dynamics

Rehabilitation and conservation programmes for historic buildings and urban landscapes necessarily need integrated strategies that include scientific research into development models designed from the bottom up, based on the particular realities of the respective territories, the coordinating role of the local public administration, and the participation of the agents and promoters active in the territory. This model has proved decisive, especially in the case study of Mértola [23], where it enabled the guaranteeing of a line of continuity based on valuing heritage as a factor in local development at least partially immune to the cycles of changing governments and public funding programmes for the sector.

In general, in inland settlements, the inner-wall urban neighbourhoods continue to experience abandonment with a large proportion of their buildings standing vacant, leading to a dual peripheral status: they become peripheral inside their cities, which are themselves

located in a peripheral territory on the national scale. In the case of Mértola, this trend has continued despite local programmes to encourage private individuals to rehabilitate, the significant number of properties rehabilitated by the municipality itself (as part of its public housing policy) and the growing number of buildings more recently converted to tourist accommodation. Although these situations differ greatly from the realities prevailing in the main urban centres on the coast, the debate ongoing around housing and cultural tourism-related problems on a national scale, or the tendency to replicate and generalise the discourses and solutions put forward for high-density areas in the interior of the country, may produce perverse effects. Recently, the set of measures approved by the Portuguese government for the housing sector differentiated the low-density regions in the interior of the country specifically in terms of restrictions on tourist accommodation [24]. It provides a positive differentiation, as the limitations set to the large metropolitan areas do not apply in inland territories. This type of decision, as any that takes the territorial disparity into consideration, is to be praised.

Nonetheless, some initiatives, programmes or research projects underway in other contexts, particularly in Southern Europe, are broadening the reflection on the range of integrated strategies for the rehabilitation of inner cities [25], considering, for example the need for the improvement of site resilience and mitigation of the impact of globalisation, considering the importance of settlements in relation to the landscape and economic and environmental sustainability, including agri-food excellence [26]; the urgency of cooperative processes for the rehabilitation of houses in urban centres of ageing populations with little investment capacity [27] or the possibility of establishing regional or national networks associated with traditional construction, aimed at bringing masters and companies together on the one hand, and owners, technicians or institutions on the other [28], with importance in the rehabilitation processes of these urban centres.

The cultural tourism sector remains a positive factor impacting the demographic and heritage conservation of medium and small inland towns. However, any strategy for the inner-wall historic areas of these urban centres should focus on the rehabilitation of buildings for permanent housing, combining economic and fiscal support measures with technical support instruments for the designing and carrying out of such projects. The ongoing situation of urban centre abandonment only reiterates the importance of proximity policies, the participation of various actors (from universities to communities) and the central role of local government, seeking to integrate research and knowledge about the transformation of the architectonical environment (typological processes and construction techniques) as a design tool; interventions with demonstrative effects within the framework of public housing policies and the (re)integration of certain collective use structures; significant encouragement (in the various aforementioned areas) for private sector intervention in permanent housing, and all associated with the dissemination of the best practices resulting from such processes.

**Author Contributions:** Conceptualization, A.C.R. and M.R.C.; methodology, A.C.R. and M.R.C.; writing—original draft preparation, A.C.R.; writing—review and editing, M.R.C. All the authors have read and agreed to the published version of the manuscript. All authors have read and agreed to the published version of the manuscript.

**Funding:** The research studies leading to this article were financed by national funding through the FCT—Fundação para a Ciência e a Tecnologia, I.P. (Foundation for Science and Technology, I.P.), within the framework of the projects SFRH/BD/116130/2016, UIDB/00281/2020 and UIDP/00281/2020—CEAACP.

**Data Availability Statement:** No new data were created or analyzed in this study. Data sharing is not applicable to this article.

**Conflicts of Interest:** The authors declare no conflict of interest. The funders had no role in the design of the study; in the collection, analyses, or interpretation of data; in the writing of the manuscript; or in the decision to publish the results.

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
