# Peer review of "The Contribution of Typological Studies to the Integrated Rehabilitation of Traditional Buildings: Heritage Enhancement of Urban Centres in Inner Alentejo, Portugal"

_2673-8945, doi:10.3390/architecture4010004_

Round 1
Reviewer 1 Report
Comments and Suggestions for Authors
Comments and Suggestions for Authors
Researchers should consider the following research notes:
The manuscript notion is unique and pertains to an important, influential, and vital topic, yet certain observations must be made:
but some considerations must be made:
Abstract: It is necessary to clarify the study methodology, the most important findings, and the approach.
Keywords: The number of keywords is adequate.
Introduction: The number of paragraphs is limited, and a theoretical framework of concepts connected to rehabilitation processes is required. At the end of the introduction, it should also explain the significance of the research, its problem, the research goal, and the research gap. I recommend using the resources listed below. 10.1007/978-3-030-74482-3_16
Materials and Methods: The methodology is straightforward.
Results and Discussion Appropriate and optimizes the research's significance.
Figures: You must give scientific references for any figures.
References: There aren't many of them, and the most of them are hidden, such as references 15, 17, and others.
Comments on the Quality of English Languageno comment
Author Response
Thank you for your comments. The changes made are addressed in the table attached.

Reviewer 2 Report
Comments and Suggestions for Authors
This paper discusses the importance of typological studies in rehabilitating traditional buildings in Inner Alentejo, Portugal, emphasising the need to understand the evolution of traditional architecture over history, as well as traditional construction techniques and materials. It also highlights the impact of demographic and tourist dynamics on rehabilitation and the need for integrated strategies and local government involvement in sustainable development and heritage conservation.
The manuscript is well-crafted, incorporating compelling on-site investigations and comprehensive documentation. Nevertheless, several aspects could be enhanced to enrich its scholarly contribution. I would recommend minor revisions.
· While the initial emphasis is on the issue of depopulation in Alentejo, the documented transformations primarily reflected housing expansions. It would be beneficial to elucidate how an analysis of housing expansion records could effectively tackle the regional challenge of depopulation.
· The paper's emphasis on understanding historical context is commendable. The section entitled ‘3. Results’ examines specific cases, such as those in Estremoz and Mértola. However, these focused cases would greatly benefit from expanded historical contextualisation to provide a deeper understanding of the architectural transformations discussed.
· The critique of current sustainable tourism, particularly its inapplicability to small inland towns or cities, prompts the question of a recommended strategy. Clarifying this suggested approach and distinctly outlining its divergence from existing practices would further strengthen the paper’s argument.
· While the inclusion of photos and drawings is commendable, it is essential to accompany these visuals with proper credit lines, acknowledging the sources of the figures employed.
· Figure 3 effectively illustrates evolution; however, Figures 4 and 5 lack explicit highlighting of changes or the provision of sections/plans before and after the transformations. Supplementing these figures with such information, if available, would enhance their utility.
· To augment the paper’s scholarly foundation, it is advisable to incorporate a comprehensive literature review. For instance, this review could elucidate the research gap that this paper intends to fill within the broader landscape of studies pertaining to architectural evolution.
In summary, this paper presents a commendable analysis of typological studies in the rehabilitation of traditional buildings in Inner Alentejo. Addressing the suggested enhancements would undoubtedly elevate its scholarly value and enrich its contribution.
Author Response

(The authors gave the same response as above.)

Reviewer 3 Report
Comments and Suggestions for Authors
The paper focuses on the depopulation crisis that is affecting the urban centers of inland Alentejo (Southern Portugal), and that is leading to a rapid deterioration of the traditional architectural heritage of this area. Traditional housing is often perceived as failing in terms of the living conditions provided, thus the abandonment and the rejection of the traditional house in the historic centers puts the conservation of built historic areas at risk. The authors propose a typological analysis of the traditional architecture of the inner Alentejo as a strategical contribution to the rehabilitation and preservation of this heritage. The study is based on the analysis of a large database and lead to the identification of the architectural and typological characteristics of the traditional residential architecture of this area. It also focuses on the analysis of the transformation processes conducting to the identification of the transformation patterns that responded to changing social, economic, and historical circumstances inAlentejo, as well as the current transformation trends. Finally, the authors propose an interesting insight on the rehabilitation in the context of inland demographic and tourist dynamics, underlining the importance of the rehabilitation of buildings for permanent housing and the importance of proximity policies, of the central role of local government, and of the integration of research and knowledge about the transformation of the architectonical environment.
The work provides a significant contribution for the enhancement of heritage preservation in the urban centers of the area studied and fit perfectly the journal scope. The manuscript is in general clear, well structured, the linguistic style is appropriate for a scientific article and the English language used is appropriate. The research gap is correctly and sufficiently presented. The development of the research is correct and leads through rigorous analysis to results of great relevance to the area of study considered. The conclusions present a series of interesting thoughts and pertinent considerations that can be applied to other similar areas under a depopulation crisis and with traditional architectural heritage at risk of abandonment.
The cited references are relevant and correctly introduced in the text. The main strength points of this study are the large database on which this research is based (A total of 77 examples of houses from Mértola, other 507 cases from the other four Alentejan cities – Estremoz, Borba, Moura and Serpa) and the rigorous methodology applied for documentation (up to 390 buildings documented in loco, most of them with direct architectural surveys, the others documented with archive information analysis, as documents from the construction registers maintained by municipalities, and inhabitant testimonies). The large database used is evidence of the good quality of the results obtained. It would be of great interest to make this database accessible, for example on the Internet, to other researchers who are studying the same subject in other geographical areas.
There only few points that, in my opinion, request a more comprehensive view: the introduction and contextualization of the research topic.
1. The abandonment of the inner urban settlement and of the historic centers, as well as the rejection of the traditional houses with the consequent risks for the preservation of this heritage in a common problem of many other European rural areas. The south rural areas of Italy or in the inner territories of Spain (known as the “España vaciada”) suffer for instance the same problems. The introduction of some of these cases (with citations), would help to contextualize the research topic in a more international and generalized context.
2. The authors cite the importance of the “Identifying and sharing of exemplary architectural rehabilitation projects as a fundamental contribution to this context” (lines 39-40). They also remark that “In general, urban centers lack an analytical document capable of framing an intervention in a specific building, evaluating its importance in the context of the history of domestic architecture” (lines 232-234). I suggest reviewing and citing some of the following examples in which the study of traditional architecture has already led to the formulation of recommendations and guidelines for the preservation of traditional architecture:
- AA. VV. (2020) 3D Past. Guidelines and strategies for maintenance of vernacular architecture in World Heritage sites. ISBN: 978-84-18514-09-8 (https://resarquitectura.blogs.upv.es/?page_id=19706).
- AA. VV. (2014) Versus. Lessons from vernacular heritage to sustainable architecture (https://resarquitectura.blogs.upv.es/?page_id=4883). ISBN: 978-2-906901-78-0.
- Research Projects “Versus Plus” https://versus-people.webs.upv.es/.
- Research Project “SOSTierra” https://sostierra.blogs.upv.es/
Finally, I suggest revising a series of details in order to improve the quality of the paper:
- Lines 55-56. I suggest improving the caption of the Figure 1: what want the authors show in these pictures? Furthermore, Figure 1 is not referenced in the manuscript, it should be referenced has the others pictures of the article.
- Line 99. The Figure 2 can be improved. The map, despite showing coordinates, is not sufficient to graphically locate the area under study. I suggest adding to one side of the image a map of Portugal, with the identification of Alentejo and to highlight the study area.
- Line 142: I suggest inserting a metric scale in the Figure 3.
- Line 144: I suggest inserting a metric scale in the Figure 4.
- Line 186: I suggest inserting a small arrow to indicate the main access to each building. In some cases, the main façade and main access are clear, but in other cases in which there are more accesses (from several sides), they are not clear. The use of an arrow helps to identify them graphically and clearly.
- Lines 226-227. I suggest improving the caption of the Figure 6: what want the authors show and communicate to the reader with these pictures? Furthermore, Figure 6 is not referenced in the manuscript, it should be referenced as the others pictures of the article.
- Line 314. I do not agree with the authors' definition of "positive discrimination": the term discrimination, even if accompanied by the adjective positive (for inner urban centers), evokes negative consequences for large urban centers. However, this type of policy, which the authors also consider laudatory, also has positive effects for major urban centers. The problem of mass tourism in large historic cities, the transformation of historic centers into tourist accommodations and the loss of authenticity of central heritage areas are currently the subject of debate in many countries. The government of several cities, such as Florence in Italy, is acting to limit mass tourism and return historic centers, at least partially, to the citizens. I suggest that the authors reflect on the definition of "discrimination" and, if they consider it appropriate, modify it or provide a clarification in this regard.
Author Response

(The authors gave the same response as above.)

Round 2
Reviewer 1 Report
Comments and Suggestions for Authors
thank you for you responses
Comments on the Quality of English Languagenothing needed